# Relationship between stress hyperglycemia ratio of one-year mortality in patients with heart failure: Analysis of the MIMIC-IV database

**Yimeng Wang**[☉], **Wei Xu**[☉], **Jingyang Wang, Yuyuan Shu, Yinjing Xin, Yanmin Yang**[iD]*

Emergency and Critical Care Center, Fuwai Hospital, National Center for Cardiovascular Diseases of China, Chinese Academy of Medical Sciences and Peking Union Medical College, Beijing, China

[☉] These authors contributed equally to this work.

* yyminfuwai@163.com

## Abstract

### Background

Stress Hyperglycemia Ratio (SHR) has been confirmed to be a predictor for adverse outcomes in cardiovascular diseases in recent years. However, the impact of SHR on one-year mortality in patients diagnosed with heart failure (HF) is still unclear. This study aims to explore the relationship between SHR and one-year mortality in HF patients, both complicated with and without diabetes mellitus (DM).

### Methods

This study enrolled 3747 patients with HF from the Medical Information Mart for Intensive Care (MIMIC-IV) database. 1865 patients were set into the group of lower SHR (SHR < 0.964) and 1882 patients were in the higher group (SHR ≥ 0.964). The primary endpoint was one-year mortality.

### Results

The mean age of the total study population was 69 ± 13, and 1530 (40.8%) of them were female. Finally, 188 (5.0%) patients died in the hospital and 766 (20.4%) patients died during a one-year follow-up. Patients in the higher SHR group had a longer hospital stay (2.7% vs. 2.4%, $p < 0.001$) and higher in-hospital mortality (8 vs. 7, $p < 0.001$) than those in the lower group. The Kaplan–Meier curves also show that higher SHR is associated with an elevated risk of one-year mortality in patients with HF, both in the DM and non-DM groups (all log-rank $p < 0.0001$). As a continuous variable, SHR was an independent predictor for one-year mortality [hazard ratio (HR), 2.893; 95% confidence interval (CI), 2.198–3.808]. Elevated SHR was significantly associated with higher risk of one-year mortality in patients with (HR, 1.499; 95% CI, 1.104–2.036) and without DM (HR, 1.300; 95% CI, 1.096–1.542), consistently. The

**Data availability statement:** Data could be downloaded from https://mimic.mit.edu/docs/iv/.

**Funding:** This research article was supported by National Clinical Medical Research Center for Cardiovascular Diseases (NCRC2020015) and High-Level Hospital Clinical Research Funding (2022-GSP-GG-26). The funders had no role in study design, data collection and analysis, decision to publish, or preparation of the manuscript.

**Competing interests:** The authors have declared that no competing interests exist.

RCS curve shows a gradual increase in the probability of one-year mortality as the value of SHR increases for HF patients.

## Conclusion

Our findings indicated that a higher level of SHR was associated with elevated one-year mortality in HF patients both with and without DM, suggesting that SHR is a promising stratification indicator for predicting the risk of death in patients with HF.

## Introduction

Heart failure (HF) is a prevalent cardiovascular disease, influencing 1%−3% of the general adult demographic and showing an incremental trend in recent years. The heavy burden of HF leads to a high cost of medication and a poor prognosis. The mortality rate among these patients is relatively high, with one-year mortality ranging from 15%−30% [1], underscoring the critical need for an indicator that can predict subsequent adverse events effectively and simply. Based on the symptoms, HF could be classified as acute HF and chronic HF [2]

Recent studies have declared that metabolic conditions are closely associated with the prognosis of HF [3–5]. Furthermore, insulin resistance (IR) and HF are common coexisting conditions, even among individuals without diabetes mellitus (DM) [6,7]. Studies have indicated that IR can lead to a regression in New York Heart Association (NYHA) functional class among HF patients [8]. Moreover, IR could also contribute to the progression of HF [9].

Stress hyperglycemia ratio (SHR) is an index calculated based on blood glucose and hemoglobin A1c levels, which may better reflect stress-induced hyperglycemia compared to blood glucose alone [10]. Higher SHR levels are associated with adverse events, including all-cause mortality, diabetes mellitus-associated mortality, ventricular remodeling, and other poor prognoses in various diseases [11,12], including acute myocardial infarction [13,14], coronary artery diseases [15], stroke [16,17], spontaneous intracerebral hemorrhage [18], and in patients with acute decompensated HF complicated with DM [19]. A published study has also reported that SHR is a risk factor for left ventricular systolic dysfunction [20], which is one of the most common characteristics of HF. Additionally, patients without DM may experience adverse events related to IR. Therefore, the current body of evidence aims to explore the relationship between SHR and one-year mortality among patients with both chronic HF and acute HF, as well as whether DM status will influence this relationship in patients with HF.

## Materials and methods

### Data source

This study was based on data from the Medical Information Mart for Intensive Care-IV (MIMIC-IV) database (version 2.2) and approved with Declaration of Helsinki. Data could be downloaded from https://mimic.mit.edu/docs/iv/.Since the data was

analyzed anonymously, the requirement for informed consent was waived. Author's account number is 12770107. Ethics, Consent to Participate, and Consent to Publish declarations: not applicable.

### Study population

Patients diagnosed with HF in MIMIC-IV database based on International Classification of Diseases (ICD)-9 and ICD-10 were retrospectively collected in our study; details of the ICD code were summarized in supplementary material S1 Table in S1 File. Patients were all hospitalized and only the first hospitalization diagnosed with HF were included in our study. We excluded patients who met the following criteria: patients without a record of glucose (n = 808) or hemoglobin A1c (HbA1c) were excluded (n = 16983). Patients aged less than 18 were excluded (n = 0). Overall, the final analysis included a total of 3747 patients.

### Data extraction, definitions, and calculation of SHR

Demographic characteristics, vital signs, medical history, laboratory measurements, and treatment were extracted using pgAdmin (4v7). Two researchers (YM Wang and W Xu) have checked the accuracy of the data extraction. Medical history was extracted based on ICD-9 or ICD-10. SHR was calculated by the formula included glucose and HbA1c as follows: plasma glucose (mg/dL)/(28.7 × HbA1c (%) −46.7) [21].

### Study endpoint

The primary endpoint was defined as 1-year mortality for patients from the MIMIC-IV database all followed up for at least one year.

### Statistical analysis

Continuous variables are presented as mean ± standard deviation (SD) or as median with lower and upper quartiles and tested by using the Wilcoxon-Mann-Whitney or t test, while categorical variables are presented as counts and percentages and tested with the χ2 test. Participants were categorized into two groups based on the restricted cubic spline (RCS), with 0.964 set as the turning point. One-year mortality rate was estimated by Kaplan-Meier curves, performed in total participants and in patients with and without DM. A multivariate Cox proportional-hazards model was used to adjust for confounding factors and identify factors associated with one-year mortality events. Details of adjusted confounding factors and results presented as a hazard ratio (HR) with a 95% CI were presented in Table 2. The non-linear relation between SHR and one-year mortality was illustrated with RCS curve. All the analyses were performed using software packages SPSS (version 25.0, IBM Corporation, New York, NY, USA), R (4.3.1, R Project for Statistical Computing, Vienna, Austria) and Adobe Illustrator (Adobe Inc., Mountain View, CA, USA). All statistical tests were two-sided and a value of P < 0.05 was considered significant.

### Results

Our study consisted of 3747 patients diagnosed with HF, 1865 patients were set into the group of lower SHR and 1882 patients were in the higher group according to RCS curve (Fig 1). Flow chart was presented in supplementary S1 Fig. Baseline characteristics were shown in Table 1. The mean age was 69 ± 13and 1530 (40.8%) of them were female. 1865 patients were set into the group of lower SHR and 1882 patients were in the higher group. Higher SHR groups were more likely to be associated with increasing age, a medical history of ACS, hypertension, sepsis, and tumor. Dopamine/Dobutamine and thiazide diuretic were more preferred in patients with higher SHR, while ACEI/ARB (Angiotensin-Converting Enzyme Inhibitor/angiotensin receptor blockers) and digoxin were less likely to be used in those patients. Clinical outcomes were presented in Table 2. Longer LOS (length of stay) was detected in patients with higher SHR; however as for los in ER (emergency room), these patients showed a shorter stay time.

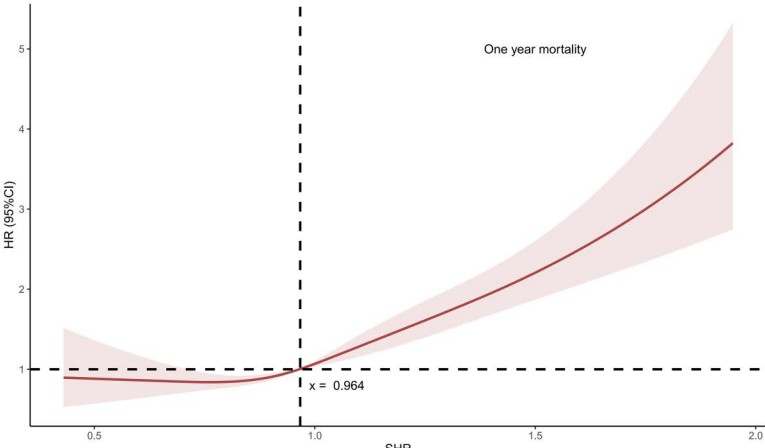

**Fig 1. Restricted cubic spline (RCS) of the overall population.** SHR: Stress hyperglycemia ratio.

### The association of SHR and one-year mortality in patients with and without DM

188 (5.0%) patients died at hospital and 767 (20.4%) patients died during one-year follow-up. Kaplan-Meier curves grouped by SHR level in total patients, patients with DM, and patients without DM for one-year mortality and in-hospital mortality were shown in Figs 2 and 3a) and b). Patients with higher SHR presented a higher one-year mortality in overall participants ($p < 0.0001$) and both patients with ($p < 0.0001$) and without DM ($p < 0.0001$).

Multivariate Cox analysis indicated that higher SHR was a risk factor as a continuous variable in the overall population (HR, 3.385; 95%CI, 2.596–4.415). After turning SHR into a categorical variable and adjusting for other risk factors, SHR (HR, 1.415; 95%CI, 1.168–1.713) has been confirmed to be an independent risk factor of one-year mortality in patients with HF. Elevated SHR represented a higher risk of one-year mortality in patients with (HR, 1.881; 95%CI, 1.244–2.846) and without DM (HR, 1.278; 95%CI, 1.025–1.593), consistently (Table 3). RCS curves showed a gradual increase in the probability of one-year mortality as the value of SHR increases for HF patients in the DM and non-DM groups (Fig 4a) and b).

### Subgroup analysis of one-year mortality

Kaplan-Meier curves grouped by SHR level in patients with AHF ($p < 0.0001$) and CHF ($p < 0.0001$) showed higher cumulative one-year-mortality events in S2 Fig a) and b). One-year mortality was consistently higher in the elevated SHR group both in patients with AHF (Log-rank $p < 0.0001$) and CHF (Log-rank $p < 0.0001$). RCS curves based on continuous SHR in patients with AHF and CHF were shown in S3 Fig a) and b).

Additionally, interaction analysis showed no statistically significant differences in all subgroup analyses, demonstrating similar risk tendency of elevated SHR among the subgroup population (Fig 5).

### Sensitivity analysis

We conducted a multivariate Cox analysis among patients with systemic inflammatory response syndrome (SIRS) scores and the simplified acute physiology score (SAPS) II. The impact of SHR on one-year mortality remained consistent even after adjusting for the SIRS and SAPS II scores. (S1 Table in S1 File).

### Discussion

In this present study, we examined the predictive value of SHR for the risk of one-year mortality among HF patients from MIMIC-IV database, regardless of the status of DM. Our findings demonstrate that a higher SHR is correlated with a

**Table 1. Baseline characteristics.**

| Variables | Total (n = 3747) | SHR < 0.964 (n = 1865) | SHR ≥ 0.964 (n = 1882) | P value |
|---|---|---|---|---|
| Age | 69 ± 13 | 69 ± 13 | 70 ± 13 | 0.001 |
| Sex (female) | 1530 (40.8%) | 746 (48.8%) | 784 (51.2%) | 0.302 |
| ER admission | 1485 (39.6%) | 765 (41.0%) | 720 (38.3%) | 0.084 |
| Race | | | | <0.001 |
| Asian | 76 (2.0%) | 37 (2.0%) | 39 (2.1%) | |
| White | 2500 (66.7%) | 1209 (64.8%) | 1291 (68.6%) | |
| Black | 404 (10.8%) | 246 (13.2%) | 158 (8.4%) | |
| Hispanic | 145 (3.9%) | 79 (4.2%) | 66 (3.5%) | |
| Other | 622 (16.6%) | 294 (15.8%) | 328 (17.4%) | |
| **Medical history** | | | | |
| ACS | 737 (19.7%) | 312 (16.7%) | 425 (57.7%) | <0.001 |
| AF | 1634 (43.6%) | 801 (42.9%) | 833 (44.3%) | 0.418 |
| CAD | 1098 (29.3%) | 581 (31.2%) | 517 (27.5%) | 0.013 |
| CKD | 1117 (29.8%) | 581 (31.2%) | 536 (28.5%) | 0.074 |
| DM | 911 (24.3%) | 472 (25.3%) | 439 (23.3%) | 0.157 |
| HT | 1174 (31.3%) | 614 (32.9%) | 560 (29.8%) | 0.037 |
| Old MI | 504 (13.5%) | 261 (14.0%) | 243 (12.9%) | 0.331 |
| Sepsis | 31 (1.7%) | 69 (3.7%) | 100 (2.7%) | <0.001 |
| Tumor | 44 (2.4%) | 88 (4.7%) | 132 (3.5%) | <0.001 |
| **Medicine** | | | | |
| ACEI/ARB | 2247 (60.0%) | 1173 (62.9%) | 1074 (57.1%) | <0.001 |
| ARNI | 29 (0.8%) | 12 (0.6%) | 17 (0.9%) | 0.364 |
| $\beta$-blocker | 3361 (89.7%) | 1667 (89.4%) | 1694 (90.0%) | 0.528 |
| Digoxin | 325 (8.7%) | 184 (9.9%) | 141 (7.5%) | 0.010 |
| MRA | 398 (10.6%) | 200 (10.7%) | 198 (10.5%) | 0.840 |
| Dopamine/ Dobutamine | 255 (6.8%) | 85 (4.6%) | 170 (9.0%) | <0.001 |
| Milirinone | 283 (7.6%) | 138 (7.4%) | 145 (7.7%) | 0.724 |
| Diuretic | | | | |
| Loop | 3137 (83.7%) | 1540 (82.6%) | 1597 (84.9%) | 0.058 |
| Thiazine | 621 (16.6%) | 282 (15.1%) | 339 (18.0%) | 0.017 |
| **Laboratory test** | | | | |
| $Cl^-$ | 101.31 ± 4.36 | 101.39 ± 4.08 | 101.24 ± 4.62 | 0.285 |
| $K^+$ | 4.20 ± 0.36 | 4.20 ± 0.35 | 4.19 ± 0.36 | 0.235 |
| RBC | 3.70 ± 0.69 | 3.80 ± 0.69 | 3.59 ± 0.67 | <0.001 |
| Hemoglobin | 10.90 ± 1.95 | 11.11 ± 1.97 | 10.70 ± 1.92 | <0.001 |
| WBC | 9.34(7.30,11.79) | 8.80(6.96,11.26) | 9.80(7.69,12.28) | <0.001 |
| PLT | 206.00 (162.67,262.75) | 208.85 (165.23,265.00) | 203.66 (159.82,261.33) | 0.039 |

Abbreviation: ACS, acute coronary syndrome; ACEI: Angiotensin-Converting Enzyme Inhibitor; ARB: angiotensin receptor blockers; ARNI: Angiotensin receptor neprilysin inhibitor; CAD: coronary artery disease; ER LOS: length of stay in emergency room; LOS: length of stay; MRA: mineralcorticoid receptor antagonist; SHR: Stress hyperglycemia ratio; PLT: platelet; RBC: red blood cell; WBC: white blood cell.

**Table 2. Clinical outcomes grouped by SHR.**

| Variables | Total | SHR < 0.964 | SHR ≥ 0.964 | *P* value |
|---|---|---|---|---|
| Los(day) | 8 (5,14) | 7 (5,13) | 8 (5,14) | <0.001 |
| ER los(h) | 6 (4,8) | 6 (4,8) | 5 (4,8) | 0.002 |
| Deaths in hospital | 188 (5.0%) | 44 (2.4%) | 144 (7.7%) | <0.001 |
| 1-year mortality | 766 (20.4%) | 308 (16.5%) | 458 (24.3%) | <0.001 |

ER los: length of stay in emergency room; Los: length of stay.

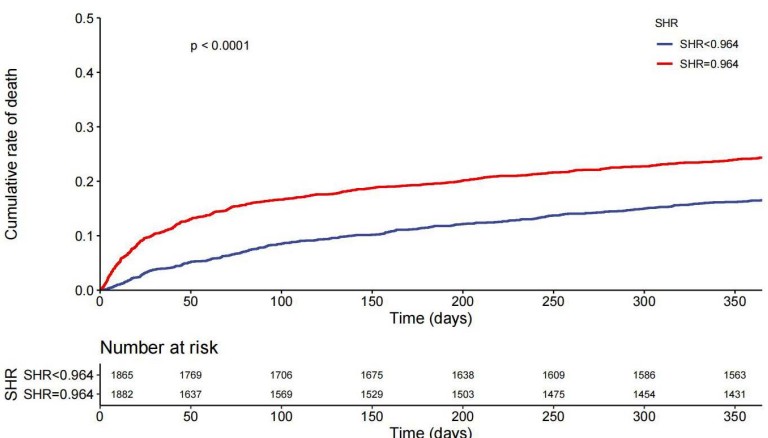

**Fig 2. One-year mortality Kaplan-Meier curves in overall participants.** SHR: Stress hyperglycemia ratio.

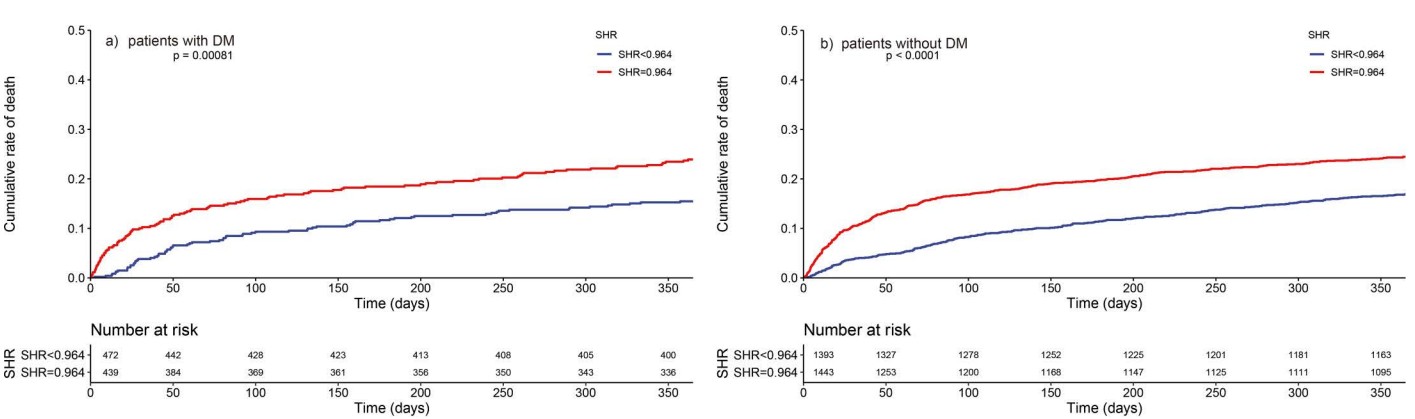

**Fig 3. One-year mortality Kaplan-Meier curves in participants a) with DM and b) without DM.** DM: Diabetes mellitus; SHR: Stress hyperglycemia ratio.

higher risk of one-year mortality in patients with HF, irrespective of the presence of DM. Specifically, the adjusted hazard ratios suggest that an elevated SHR is associated with an 88.1% increased risk of one-year mortality in HF patients with DM and a 27.8% increased risk in those without DM. Additionally, we explore the effect of SHR on one-year mortality according to the AHF and CHF groups. Our results suggest that the predictive role of SHR for one-year all-cause mortality is consistent among both AHF and CHF patients. This consistent association across different subgroups underscores

**Table 3. Cox analysis for stress hyperglycemia ratio.**

|  | HR | 95%CI |
|---|---|---|
| SHR (continuous) | 3.385 | 2.596-4.415 |
| SHR (categorical) |  |  |
| Model1 | 1.549 | 1.340-1.791 |
| Model2 | 1.478 | 1.278-1.710 |
| Model3 | 1.415 | 1.168-1.713 |
| Non-DM |  |  |
| Model1 | 1.536 | 1.302-1.813 |
| Model2 | 1.435 | 1.214-1.696 |
| Model3 | 1.278 | 1.025-1.593 |
| DM |  |  |
| Model1 | 1.568 | 1.162-2.116 |
| Model2 | 1.597 | 1.182-2.158 |
| Model3 | 1.881 | 1.244-2.846 |

Continuous SHR: adjust age, gender, race, history of ACS, HT, sepsis and tumor, ACEI/ARB, digoxin, dopamine/dobutamine, thiazine diuretic, length of stay, RBC, WBC, PLT and hemoglobin.

Model1: adjust age, gender and race; Model 2: adjust age, gender, race, history of ACS, CAD, HT, sepsis and tumor; Model 3, adjust age, gender, race, history of ACS, CAD, HT, sepsis and tumor, ACEI/ARB, digoxin, dopamine/dobutamine, thiazine diuretic, length of stay, RBC, WBC, PLT and hemoglobin.

Abbreviation: ACS, acute coronary syndrome; ACEI: Angiotensin-Converting Enzyme Inhibitor, ARB: angiotensin receptor blockers; CAD: coronary artery disease; SHR: Stress hyperglycemia ratio; PLT: platelet; RBC: red blood cell; WBC: white blood cell.

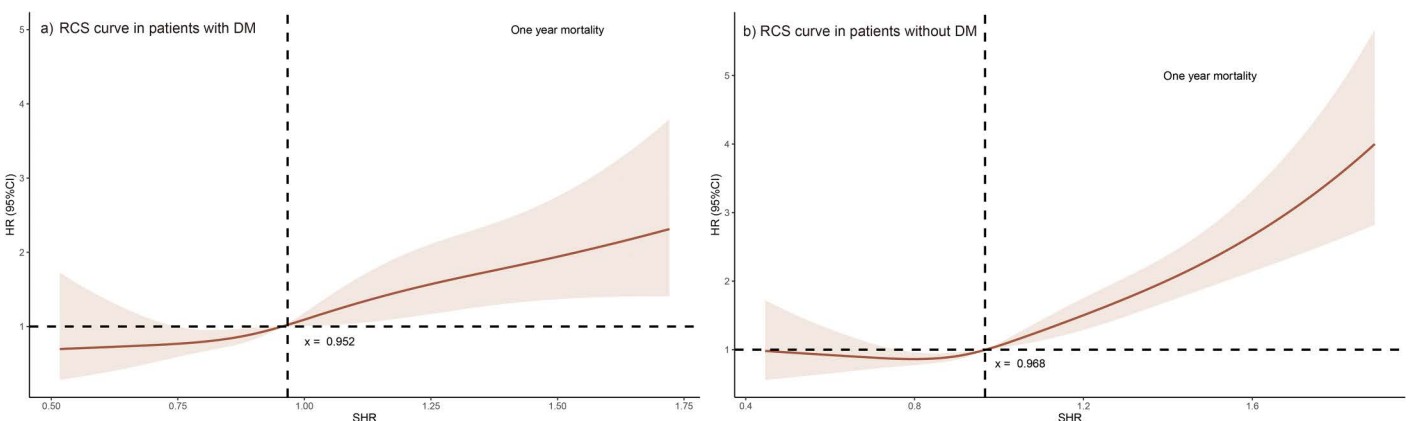

**Fig 4. Restricted cubic spline (RCS) in participants a) with DM and b) without DM.** DM: Diabetes mellitus; HR: hazard ratio; SHR: Stress hyperglycemia ratio.

the robustness of SHR as a predictor for poor prognosis in HF. The Kaplan–Meier curves also show that higher SHR is associated with an elevated risk of one-year mortality in patients with HF, both in the DM and non-DM groups. The RCS curve shows a gradual increase in the probability of one-year mortality as the value of SHR increases for HF patients with DM. Similarly, for HF patients without DM, the RCS curve also demonstrates a comparable trend. Both curves suggest a negative correlation between SHR and survival probability in HF patients with and without DM. The longer length of hospital stay observed in the high SHR group is also noteworthy. Prolonged hospitalization may indicate more severe HF or complications, consistent with the observed association of higher SHR with increased mortality risk.

| Variable | | HR(95%CI) | P value | P for interaction |
|---|---|---|---|---|
| Gender | | | | 0.896 |
| Male | | 1.575(1.289–1.925) | <0.001 | |
| Female | | 1.580(1.280–1.949) | <0.001 | |
| Race | | | | 0.496 |
| Asian | | 1.190(0.374–3.791) | 0.768 | |
| White | | 1.590(1.328–1.902) | <0.001 | |
| Black | | 1.1336(0.713–1.809) | 0.592 | |
| Hispanic | | 1.377(0.500–3.789) | 0.536 | |
| Other | | 1.845(1.327–2.566) | <0.001 | |
| DM | | | | 0.553 |
| Yes | | 1.675(1.241–2.262) | 0.001 | |
| No | | 1.522(1.288–1.797) | <0.001 | |
| CKD | | | | 0.914 |
| Yes | | 1.579(1.254–1.988) | <0.001 | |
| No | | 1.537(1.274–1.856) | <0.001 | |
| Sepsis | | | | 0.939 |
| Yes | | 1.49(0.754–2.948) | 0.252 | |
| No | | 1.578(1.359–1.831) | <0.001 | |
| HF type | | | | 0.057 |
| AHF | | 1.416(1.192–1.683) | <0.001 | |
| CHF | | 1.947(1.488–2.548) | <0.001 | |

**Fig 5. Hazard ratio of elevated SHR among the subgroup population.** HR: hazard ratio; CKD: chronic kidney disease; DM: Diabetes mellitus; SHR: Stress hyperglycemia ratio.

Extensive research has shown that stress hyperglycemia is widely recognized in patients with severe illnesses and is closely aligned with the risk of adverse events [22–24]. Higher SHR levels are associated with a 53% increased risk of sepsis [25]. The association between nonalcoholic fatty and SHR was also reported. Among this population, stress hyperglycemia is driven by the combined effects of the hypothalamic-pituitary-adrenal axis, the sympathetic-adrenal system, and pro-inflammatory cytokines [26]. Moreover, stress hyperglycemia is a marker indicating adverse clinical outcomes for patients with critical illness, particularly those with cardiovascular diseases [27–29]. SHR, proposed by Robert et al., aims to more accurately identify stress-induced hyperglycemia without being influenced by the patient's baseline blood glucose levels and has been found to effectively predict adverse outcomes in critically ill patients [29]. A multicenter study involving 5,417 acute STEMI patients demonstrated that SHR is significantly associated with the short-term mortality risk [30]. Yang et al. found that SHR has a U-shaped correlation with MACE rate and a J-shaped correlation with in-hospital cardiac death in ACS patients who underwent stent implantation, and demonstrated that SHR correlates with the long-term clinical outcomes among this cohort of patients [15]. However, in our study, no A prospective study involving 1,553 consecutive patients with acute myocardial infarction (AMI) confirmed that SHR demonstrates predictive power for a composite end-point of in-hospital mortality, pulmonary edema, and cardiogenic shock, outperforming the baseline blood glucose value at admission [31].

Aligning with earlier observations in patients with other cardiovascular diseases, our findings confirm the prognostic significance of SHR in patients with HF. The results of our study indicated that SHR was significantly related to the risk of one-year mortality for HF patients. Indeed, only a limited number of prior investigations have explored the impact of SHR on HF. A retrospective study involving 8,268 individuals with congestive HF demonstrated a U-shaped trend between the SHR and the occurrence of acute kidney injury (AKI) during hospitalization [32]. But the study did not further explore the impact of SHR on the follow-up clinical outcomes of patients with congestive HF. Zhou et al. observed a U-shaped relationship between SHR and adverse outcome events, implying that both excessively high and low SHR values are

indicative of adverse clinical outcomes for patients with HF and DM [19]. However, Zhou's study was conducted in patients with acute decompensated HF and divided 780 subjects into five groups according to SHR quintile, with a sample size of 156 cases per group, which may influence the results due to the limited sample size in each group. In our study, we expanded the study population and enrolled HF patients both with DM and without DM. Due to differences in participant enrollment, the aforementioned RCS shape was not observed in our study. However, our results indicate that among patients with HF, as the SHR value increases, the risk of one-year mortality increases correspondingly, regardless of whether they have DM or not. Moreover, we conducted subgroup analysis based on whether the type of HF was AHF or CHF, showing the consistent impact of SHR on one-year mortality in this population.

It is widely recognized that stress hyperglycemia is linked to an elevated risk of mortality and adverse clinical outcomes among patients with critical illness [24,33]. However, intensive glucose-lowering treatments may paradoxically raise the risk of death, making the blood glucose management for critically ill patients a challenge [34]. A large-scale international prospective randomized controlled study suggested that intensive blood glucose control increased the risk of death in ICU patients, possibly due to the occurrence of severe hypoglycemia [34]. A retrospective study conducted in 45,000 critically ill patients suggested that for patients without DM, a mean blood glucose level of 4.4 to 7.8 mmol/L was linked to a reduced risk of death, while patients with DM were more well tolerated, those with a mean blood glucose level of > 6.1 mmol/L had a lower mortality risk than DM patients with a mean blood glucose level of 4.4 to 6.1 mmol/L [35]. Moreover, the presence of stress hyperglycemia in patients generally suggests a more critical condition. The results of our study indicate the high level of SHR group has older age, is more likely to have ACS and CAD, has higher white blood cell levels, and lower hemoglobin levels, all of which confirm that patients with stress hyperglycemia have a more severe condition and are more likely to have other comorbidities. Therefore, for HF patients experiencing stress hyperglycemia, optimal management requires both reasonable glycemic control and rigorous management of the HF and co-existing conditions. Meanwhile, SHR levels can be influenced by factors such as sepsis, cerebrovascular diseases, cardiovascular diseases, and other critical illnesses [36]. Therefore, we adjusted for the relevant variables during the analysis, but the interpretation of SHR levels should be approached with caution. But still, nutritional status and some other diseases may also affect the level of SHR, therefore, when under this special situation, the result should be interpreted carefully [37]. The current ESC guidelines recommend using sodium-glucose transporter 2 (SGLT2) inhibitors for managing HF patients, regardless of left ventricular ejection fraction (LVEF), once the patient's condition is stable [38]. One study suggests that SGLT2 inhibitors may exert cardioprotective effects by inhibiting autophagy in cardiomyocytes, thereby improving survival rates in mice with MI accompanied by acute hyperglycemia [39]. Patients with HF and stress hyperglycemia may derive greater benefits from the treatment with SGLT2 inhibitors, but this requires further research to confirm.

To our knowledge, this is the first study to evaluate the value of SHR in both AHF and CHF patients. Additionally, we confirmed the consistency of SHR's impact on one-year mortality in both DM and non-DM patients with HF. Although SHR can be influenced by various factors such as infection, tumors, and other stress-related conditions, our results indicated that it remains a convenient index for predicting prognosis in the HF population. Given its potential prognostic value, SHR could serve as a convenient tool for risk stratification in patients with HF. Additionally, further prospective studies are still needed to explore the application of SHR in this population during treatment.

## Limitation

There are several limitations in this study. First, despite the MIMIC-IV database including individuals from various ethnic groups such as Asian, White, Black, and Hispanic, the retrospective study design and the severe disease study population from the database may limit the generalizability of the study results. Second, the study outcome may be affected by various uncollected potential factors, such as socioeconomic conditions and nutritional status of patients. Third, disease diagnoses were based on ICD codes, which may introduce bias. Additionally, further research should confirm these findings across diverse populations and investigate in greater depth the mechanisms of SHR on HF prognosis. In the future,

studies should be conducted to investigate the optimal treatment for patients with stress hyperglycemia to improve the survival rate for HF patients.

## Conclusion

Our findings indicated that higher level of SHR was associated with elevated one-year mortality in HF patients both with and without DM, suggesting that SHR is a promising stratification indicator for predicting the risk of death in patients with HF.

## Supporting information

**S1 File. S1 Table.** International Classification of Diseases (ICD) code.
(DOCX)

**S1 Fig. Flow chart of population.**
(JPG)

**S2 Fig. One-year mortality Kaplan-Meier curves in participants a) with AHF and b) with CHF.**
(JPG)

**S3 Fig. Restricted cubic spline (RCS) in participants a) with AHF and b) with CHF.**
(JPG)

## Author contributions

**Data curation:** Yimeng Wang, Wei Xu, Jingyang Wang, Yuyuan Shu, Yinjing Xin.

**Writing – original draft:** Yimeng Wang, Wei Xu.

**Writing – review & editing:** Yanmin yang.

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
