## [Decision Letter · Decision Letter 0]

Dear Dr. yang,

Thank you for submitting your manuscript to PLOS ONE. After careful consideration, we feel that it has merit but does not fully meet PLOS ONE’s publication criteria as it currently stands. Therefore, we invite you to submit a revised version of the manuscript that addresses the points raised during the review process.

We look forward to receiving your revised manuscript.

Kind regards,

Fuad Abdu

Academic Editor

PLOS ONE

“This research article was supported by National Clinical Medical Research Center for Cardiovascular Diseases (NCRC2020015) and High-Level Hospital Clinical Research Funding (2022-GSP-GG-26).

Reviewers' comments:

Reviewer's Responses to Questions

**Comments to the Author**

1. Is the manuscript technically sound, and do the data support the conclusions?

Reviewer #1: Partly

Reviewer #2: Yes

Reviewer #3: Yes

Reviewer #4: Yes

2. Has the statistical analysis been performed appropriately and rigorously?

Reviewer #1: Yes

Reviewer #2: Yes

Reviewer #3: Yes

Reviewer #4: Yes

3. Have the authors made all data underlying the findings in their manuscript fully available?

Reviewer #1: Yes

Reviewer #2: Yes

Reviewer #3: Yes

Reviewer #4: Yes

4. Is the manuscript presented in an intelligible fashion and written in standard English?

Reviewer #1: Yes

Reviewer #2: No

Reviewer #3: Yes

Reviewer #4: Yes

Reviewer #1: General Comments

This study used the MIMIC-IV database to investigate the association between the stress hyperglycemia ratio (SHR) and 1-year mortality in patients with heart failure (HF). SHR was found to be independently associated with 1-year mortality regardless of diabetes status, suggesting its potential as a prognostic indicator for patients with heart failure. However, several concerns remain that need to be addressed.

Major Comments

(1) There is no explanation for the SHR cutoff (0.964) based on ROC analysis or clinical validity. Please specify the method used to determine the cutoff (e.g., Youden index or references to previous literature).

(2) Infections (presence of sepsis), cancer, liver disease, nutritional status, and other factors may influence the outcome but may not have been included in the analysis. Consideration of these confounding factors should be added to the Discussion section.

(3) Diabetes diagnosis appears to have been based on ICD codes, but complementary definitions using measured blood glucose or HbA1c levels should also be considered. The limitations of diabetes classification using ICD codes should be mentioned in the Limitations section.

(4) Previous studies have reported U-shaped or J-shaped relationships between SHR and prognosis, but this study interprets the relationship as a linear increase in risk. Please discuss the reasons for this interpretation and explain why the shape of the RCS is consistent or inconsistent with other studies.

(5) In some patients with heart failure, particularly those with severe cases admitted to the intensive care unit (ICU), systemic inflammatory response and sepsis may occur simultaneously, and these conditions have been reported to significantly impact prognosis and pathophysiology. The involvement of sepsis-associated myocardial dysfunction is increasingly recognized as a critical factor determining the deterioration of hemodynamics. (Sci Rep. 2021;11:18823.) Considering the systemic inflammation and catabolic stress observed in heart failure and sepsis, SHR may not only reflect blood glucose fluctuations but also indicate the extent of inflammatory load. In fact, elevated SHR may serve as an alternative marker for microcirculatory dysfunction and blood flow imbalance common to heart failure and sepsis.

As mentioned earlier, findings regarding SHR in critically ill patients admitted to the ICU should be interpreted within a broader clinical framework that includes infection, inflammatory processes, cardiac dysfunction, and hemodynamic abnormalities in the context of heart failure and sepsis. Please incorporate the above content into the Discussion section.

Minor comments

(1) The consistency of labels and legends in figures and tables should also be reviewed (e.g., explicit titles for Figure 4a) and b) would be helpful).

Reviewer #2: Wang et al. Have focused on the relationship between Stress hyperglycemia ratio of one-year mortality in patients with heart failure. They found higher level of SHR was associated with elevated one year mortality in HF patients both with and without DM. The results were credible.

I have several Minor comments:

1. 1. Please indicate whether of your present study was sufficient support your results.

2. Inflammation is an important factor in blood glucose stress. Should inflammation analysis be included in this study

3. What is the difference between in-hospital mortality rate and 1-year follow-up mortality rate

4.Is there a difference in subgroups of heart failure

5.Throughout the manuscript there are a considerable number of grammar errors. To improve the readability of the manuscript, it has to undergo linguistically and grammatical changes. I would suggest using a copy-editing service for copywriting, language checking and proofreading.

Reviewer #3: Dear Authors,

Thanks for your interesting subject and work. That's an important issue in patients with heart failure, in my point of view it's better that at the end of discussion, you mention about the clinical importance of the work.

Reviewer #4: Dear author,

I have reviewed your article with great care and pleasure.

The introduction, methods, results and discussion sections of your article are written in a fluent, understandable and well-ordered manner. It was seen that the tables and figures are readable, clear and understandable.

The points that I will criticize negatively about your article are as follows:

1) The center where the ethics were obtained, the ethics date and the ethics number should be stated.

2) The introduction section is short and superficial. It would be better for your article if you explained heart failure a little more and mentioned the conditions that affect the diagnosis, treatment, prognosis and mortality of heart failure.

3) It would be good to add examples from previous studies conducted in terms of heart failure prognosis and mortality in the introduction section and increase the number of your references.

4) In your method section, you should also state how you made the diagnosis and classification of heart failure and which diagnostic tools you used.

5) Your inclusion and exclusion criteria in the method section are very simple. You should specify them in more detail.

I liked your study in general terms.

**Do you want your identity to be public for this peer review?** For information about this choice, including consent withdrawal, please see our Privacy Policy

Reviewer #1: No

Reviewer #2: No

Reviewer #3: **Yes: ** zahra khajali

Reviewer #4: **Yes: ** Azmi Eyiol

---

## [Author Response · Author response to Decision Letter 1]

22 Jun 2025

Dear Editors and Reviewers,

Thank you very much for providing us with the opportunity to revise our manuscript.

We are grateful to the editor and reviewers for their constructive comments and helpful suggestions. Responding to the critics, we have conducted additional analyses as requested. We also improved our expression in the background and discussion. We believe that our manuscript has been substantially improved and hope it is now acceptable for the Thrombosis Journal.

I assure you that all authors have read and approved the submission of the revised manuscript. Also, this work, containing the original research, has not been under consideration for publication elsewhere.

Please let me know if you have any further questions.

With warmest regards,

Yanmin Yang

Emergency and Critical Care Center, Fuwai Hospital, National Center for Cardiovascular Diseases of China, Chinese Academy of Medical Sciences and Peking Union Medical College, Beijing, China

E-mail: yyminfuwai@163.com

The main corrections in the paper and the responses to the reviewer’s comments are as follows (All the text in bold font is the original comments from the editor/reviewers, and our response is in normal font):

Reviewer #1: General Comments

This study used the MIMIC-IV database to investigate the association between the stress hyperglycemia ratio (SHR) and 1-year mortality in patients with heart failure (HF). SHR was found to be independently associated with 1-year mortality regardless of diabetes status, suggesting its potential as a prognostic indicator for patients with heart failure. However, several concerns remain that need to be addressed.

Major Comments

(1) There is no explanation for the SHR cutoff (0.964) based on ROC analysis or clinical validity. Please specify the method used to determine the cutoff (e.g., Youden index or references to previous literature).

Thank you for your insightful comment. We have added the relevant information in the methods section to make it easier for readers to understand. (page 4, line 87-88) First, we used restricted cubic spline (RCS) to analyze the data and observe the trend. When the stress hyperglycemia ratio (SHR) exceeded 0.964, the one-year mortality risk increased with the rising SHR, with a hazard ratio greater than 1 (Fig. 1). Therefore, we set 0.964 as the cutoff for group classification, which could offer valuable insights for clinical practice in the future. Moreover, we have analyzed the cutoff (0.964) on ROC, the UAC (0.560, 95% CI: 0.537-0.583). Although the prediction value of this cutoff was moderately good, it also presented its efficiency in mortality prediction.

(2) Infections (presence of sepsis), cancer, liver disease, nutritional status, and other factors may influence the outcome, but may not have been included in the analysis. Consideration of these confounding factors should be added to the Discussion section.

Thank you for your suggestion to include the variables that might influence SHR and the interpretation of results.

We have added the relevant references in the discussion section. (page 14, line 254-259; page 14, line 276)

(3) Diabetes diagnosis appears to have been based on ICD codes, but complementary definitions using measured blood glucose or HbA1c levels should also be considered. The limitations of diabetes classification using ICD codes should be mentioned in the Limitations section.

Thanks for your comments. We addressed the potential bias introduced in this study due to classification based on ICD codes in the limitations section(page 14, line 277)

(4) Previous studies have reported U-shaped or J-shaped relationships between SHR and prognosis, but this study interprets the relationship as a linear increase in risk. Please discuss the reasons for this interpretation and explain why the shape of the RCS is consistent or inconsistent with other studies.

Thanks for your valuable comments. We have discussed the differences between our study and other published studies. �page 13, line 221-232� In brief, we hypothesized that the differences arose due to the distinct populations.

(5) In some patients with heart failure, particularly those with severe cases admitted to the intensive care unit (ICU), systemic inflammatory response and sepsis may occur simultaneously, and these conditions have been reported to significantly impact prognosis and pathophysiology. The involvement of sepsis-associated myocardial dysfunction is increasingly recognized as a critical factor determining the deterioration of hemodynamics. (Sci Rep. 2021;11:18823.) Considering the systemic inflammation and catabolic stress observed in heart failure and sepsis, SHR may not only reflect blood glucose fluctuations but also indicate the extent of inflammatory load. In fact, elevated SHR may serve as an alternative marker for microcirculatory dysfunction and blood flow imbalance common to heart failure and sepsis.

Thanks for your suggestions. We added sepsis and tumor status to the baseline characteristics and adjusted for them in the multivariate Cox analysis (Table 1 and Table 3). Furthermore, we performed a sensitivity analysis based on patients with systemic inflammatory response syndrome (SIRS) and the simplified acute physiology score (SAPS) II to exclude the potential influence of inflammation and sepsis on the results. (supplementary table S1)

We also added this content to the discussion section. (page 12; line 200-201; page 14, line 254-256)

As mentioned earlier, findings regarding SHR in critically ill patients admitted to the ICU should be interpreted within a broader clinical framework that includes infection, inflammatory processes, cardiac dysfunction, and hemodynamic abnormalities in the context of heart failure and sepsis. Please incorporate the above content into the Discussion section.

Thank you for your suggestion. We have added the relationship between SHR and other critical illnesses in the discussion section to remind readers that the interpretation of SHR should be approached with caution. (page 14, line 254-256)

Minor comments

(1) The consistency of labels and legends in figures and tables should also be reviewed (e.g., explicit titles for Figure 4a) and b) would be helpful).

Thank you for your suggestions. We have reviewed the consistency of labels and legends in the figures and tables to ensure accuracy.

Reviewer #2: Wang et al. Have focused on the relationship between Stress hyperglycemia ratio of one-year mortality in patients with heart failure. They found higher level of SHR was associated with elevated one year mortality in HF patients both with and without DM. The results were credible.

I have several Minor comments:

1. 1. Please indicate whether of your present study was sufficient support your results.

Thanks for your constructive suggestion. We have validated the findings of our study in discussion section. (page 14, line 266-270)

2. Inflammation is an important factor in blood glucose stress. Should inflammation analysis be included in this study

Thanks for your suggestions. Regarding the methods, we extracted variables reflecting the inflammatory status, such as the Sequential Organ Failure Assessment (SOFA) score, the Systemic Inflammatory Response Syndrome (SIRS) score, and the level of C-reactive protein. However, due to missing values exceeding 50%, we did not include these variables in our study. Nevertheless, inspired by your suggestion, we added sepsis and tumor status to the baseline characteristics and adjusted for them in the multivariate Cox analysis (Table 1 and Table 3).

3. What is the difference between in-hospital mortality rate and 1-year follow-up mortality rate

Thank you for your valuable comment regarding differences between in-hospital mortality and 1-year mortality. We assumed that in-hospital mortality was associated with short-term adverse events, representing deaths that occurred during the admission period, while 1-year mortality was linked to long-term adverse events. Therefore, we presented both results. (Table 2)

4. Is there a difference in subgroups of heart failure

Thanks for your comment regarding the statistical analysis. We have added the subgroup analysis. There were no interaction differences between acute heart failure and chronic heart failure (Figure 5).

5.Throughout the manuscript there are a considerable number of grammar errors. To improve the readability of the manuscript, it has to undergo linguistically and grammatical changes. I would suggest using a copy-editing service for copywriting, language checking and proofreading.

Thank you for your constructive suggestions. We have revised the language and grammar throughout the entire text by a native speaker.

Reviewer #3: Dear Authors,

Thanks for your interesting subject and work. That's an important issue in patients with heart failure, in my point of view it's better that at the end of discussion, you mention about the clinical importance of the work.

Thanks for your constructive comments. At the end of the discussion, we added the potential usage of SHR in clinical practice. (page 14,line 266-270)

Reviewer #4: Dear author,

I have reviewed your article with great care and pleasure.

The introduction, methods, results and discussion sections of your article are written in a fluent, understandable and well-ordered manner. It was seen that the tables and figures are readable, clear and understandable.

The points that I will criticize negatively about your article are as follows:

1) The center where the ethics were obtained, the ethics date and the ethics number should be stated.

Thanks for your valuable comments. Since the data was extracted from a publicly established database and approved in accordance with the Declaration of Helsinki. Ethics approval dates and numbers were not applicable. (page 3, line 62-63, 65-66)

2) The introduction section is short and superficial. It would be better for your article if you explained heart failure a little more and mentioned the conditions that affect the diagnosis, treatment, prognosis and mortality of heart failure.

Thank you for your comments on the introduction section. We have revised this part and refined the background information related to heart failure. (page 3, line 40-41,55-56)

3) It would be good to add examples from previous studies conducted in terms of heart failure prognosis and mortality in the introduction section and increase the number of your references.

Thank you for your constructive suggestion. We have revised the introduction section and increased the number of references.

(W. Zhang, Y. Zhang, J. Tang, X. Wang, C. Meng, J. Wu, et al. The changing landscape of heart failure drug clinical trials in china, 2013-2023[J]. Drug Des Devel Ther, 2025, 19: 2597-2608.

[3]M. Saotome, T. Ikoma, P. Hasan, Y. Maekawa. Cardiac insulin resistance in heart failure: The role of mitochondrial dynamics[J]. Int J Mol Sci, 2019, 20(14):

U. Attanasio, V. Mercurio, S. Fazio. Insulin resistance with associated hyperinsulinemia as a cause of the development and worsening of heart failure[J]. Biomedicines, 2024, 12(12):

J. Zhu, W. Liu, L. Chen, B. Liu. Stress hyperglycemia ratio: A novel predictor of left ventricular dysfunction in peripartum cardiomyopathy[J]. J Matern Fetal Neonatal Med, 2025, 38(1): 2464181.)

4) In your method section, you should also state how you made the diagnosis and classification of heart failure and which diagnostic tools you used.

Thank you for your comments. We have added the criteria for the diagnosis of heart failure, which were based on the International Classification of Diseases (ICD)-9 and ICD-10, in the methods section. The details of the ICD codes and associated diseases are provided in Supplementary Table S1. We have classified heart failure into chronic and acute heart failure according to ICD codes.

5) Your inclusion and exclusion criteria in the method section are very simple. You should specify them in more detail.

Thank you for your valuable suggestions regarding the inclusion and exclusion criteria. We have provided more details on the criteria for this study (page 4, line 72-74). We did consider age as an exclusion criterion; however, since no patients were excluded based on this criterion, we chose not to mention it. (Supplementary Figure S1)

---

## [Decision Letter · Decision Letter 1]

Dear Dr. yang,

Thank you for submitting your manuscript to PLOS ONE. After careful consideration, we feel that it has merit but does not fully meet PLOS ONE’s publication criteria as it currently stands. Therefore, we invite you to submit a revised version of the manuscript that addresses the points raised during the review process.

We look forward to receiving your revised manuscript.

Kind regards,

Fuad Abdu

Academic Editor

PLOS ONE

Journal Requirements:

Reviewers' comments:

Reviewer's Responses to Questions

**Comments to the Author**

Reviewer #1: (No Response)

Reviewer #2: All comments have been addressed

Reviewer #3: All comments have been addressed

Reviewer #4: All comments have been addressed

2. Is the manuscript technically sound, and do the data support the conclusions?

Reviewer #1: Yes

Reviewer #2: Yes

Reviewer #3: Yes

Reviewer #4: Yes

3. Has the statistical analysis been performed appropriately and rigorously?

Reviewer #1: Yes

Reviewer #2: Yes

Reviewer #3: Yes

Reviewer #4: Yes

4. Have the authors made all data underlying the findings in their manuscript fully available?

Reviewer #1: Yes

Reviewer #2: Yes

Reviewer #3: Yes

Reviewer #4: Yes

5. Is the manuscript presented in an intelligible fashion and written in standard English?

Reviewer #1: Yes

Reviewer #2: Yes

Reviewer #3: Yes

Reviewer #4: Yes

Reviewer #1: Minor comments:

1) The authors' responses to the previous review were generally appropriate, and the following points have been adequately addressed:

・The rationale for the SHR cutoff was clearly stated based on RCS and ROC analysis, and the interpretation was reasonable.

・The inclusion of confounding factors such as inflammation, tumor, and sepsis in the Cox regression model, as well as sensitivity analysis using SIRS and SAPS II, is highly commendable.

・The limitations of DM and HF classification based on ICD codes are also appropriately noted.

・Differences in the shape of the RCS curve compared to previous studies are discussed in terms of differences in the study population.

・Linguistic revisions and corrections to figure labels have been made, improving the overall clarity of the paper.

However, we believe that further improvements are possible in the following areas:

2) This study focuses on the association between metabolic stress, insulin resistance, and heart failure prognosis. From the perspective of metabolic abnormalities in heart failure, the following studies may provide further insights into the significance of SHR: ESC Heart Fail. 2023;10:32-43. Heart Vessels. 2021;36:965-977. J Cardiol. 2020;75:689-696.

3) Although the somewhat limited AUC of 0.56 in the ROC analysis is briefly mentioned, I feel that a more in-depth discussion on its clinical applicability and practical significance would be beneficial.

Reviewer #2: Thank you for performing all the new experiments and the new data. This manuscript is substantially improved.

I believe it is now acceptable for publication in Plos one in its present form.

Reviewer #3: Dear Authors,

Thanks for your revision. You added clinical significance of the study at the end of discussion.

Reviewer #4: Dear author,

I saw that you have made the necessary corrections to your article. I think that your article is a candidate for publication in its current state.

**Do you want your identity to be public for this peer review?** For information about this choice, including consent withdrawal, please see our Privacy Policy

Reviewer #1: No

Reviewer #2: No

Reviewer #3: **Yes: ** zahra khajali

Reviewer #4: **Yes: ** Azmi Eyiol

---

## [Author Response · Author response to Decision Letter 2]

3 Jul 2025

Point-to-point response

Dear Editors and Reviewers,

Thank you very much for providing us with the opportunity to revise our manuscript.

We are grateful to the editor and reviewers for their constructive comments and helpful suggestions. Responding to the critics, we have conducted additional analyses as requested. We also improved our expression in background and discussion. We believe that our manuscript has been substantially improved and hope it is now acceptable for the Thrombosis Journal.

I assure you that all authors have read and approved the submission of the revised manuscript. Also, this work containing the original research has not been under consideration for publication elsewhere.

Please let me know if you have any further questions.

With warmest regards,

Yanmin Yang,

Emergency and Critical Care Center, Fuwai Hospital, National Center for Cardiovascular Diseases of China, Chinese Academy of Medical Sciences and Peking Union Medical College, Beijing, China

E-mail: yyminfuwai@163.com

The main corrections in the paper and the responses to the reviewer’s comments are as follows (All the text in bold and underlined font are the original comments from the editor/reviewers and our response is in normal font):

To reviewer 1:

This study focuses on the association between metabolic stress, insulin resistance, and heart failure prognosis. From the perspective of metabolic abnormalities in heart failure, the following studies may provide further insights into the significance of SHR: ESC Heart Fail. 2023;10:32-43. Heart Vessels. 2021;36:965-977. J Cardiol. 2020;75:689-696.

Thank you for your insightful comments. We have added the association between metabolic conditions and heart failure prognosis in the introduction section and have cited the references as per your recommendation. (page 3,line 45-46)

Although the somewhat limited AUC of 0.56 in the ROC analysis is briefly mentioned, I feel that a more in-depth discussion on its clinical applicability and practical significance would be beneficial.

Thanks for your constructive suggestions. We added a discussion on its clinical applicability and practical significance in discussion section.(Page 14, line 270-275) 

To reviewer 2:

Special thanks for reviewer 2.

To reviewer 3:

Special thanks for reviewer 3.

To reviewer 4:

Special thanks for reviewer 4.

---

## [Editor Report · Decision Letter 2]

Relationship between Stress hyperglycemia ratio of one-year mortality in patients with heart failure:analysis of the MIMIC-IV database.

PONE-D-25-21818R2

Dear Dr. yang,

We’re pleased to inform you that your manuscript has been judged scientifically suitable for publication and will be formally accepted for publication once it meets all outstanding technical requirements.

Kind regards,

Fuad Abdu

Academic Editor

PLOS ONE